# Periodontal Disease and Vitamin D Deficiency in Pregnant Women: Which Correlation with Preterm and Low-Weight Birth?

**DOI:** 10.3390/jcm10194578

**Published:** 2021-10-02

**Authors:** Martina Ferrillo, Mario Migliario, Andrea Roccuzzo, Pedro Molinero-Mourelle, Giovanni Falcicchio, Giuseppina Rosaria Umano, Federica Pezzotti, Pier Luigi Foglio Bonda, Dario Calafiore, Alessandro de Sire

**Affiliations:** 1Department of Surgical Sciences, Dental School, University of Turin, 10126 Turin, Italy; 2Department of Translational Medicine, University of Eastern Piedmont, 28100 Novara, Italy; pierluigi.fogliobonda@med.uniupo.it; 3Dental Clinic, University Hospital “Maggiore della Carità”, 28100 Novara, Italy; federica.pezzotti@maggioreosp.novara.it; 4Department of Periodontology, School of Dental Medicine, University of Bern, 3010 Bern, Switzerland; andrea.roccuzzo@zmk.unibe.ch; 5Department of Oral and Maxillofacial Surgery, Copenhagen University Hospital (Rigshospitalet), 2100 Copenhagen, Denmark; 6Department of Reconstructive Dentistry and Gerodontology, School of Dental Medicine, University of Bern, Freiburgstrasse 7, 3010 Bern, Switzerland; pedro.molineromourelle@zmk.unibe.ch; 7Department of Conservative Dentistry and Orofacial Prosthodontics, Faculty of Dentistry, Complutense University of Madrid, Plaza de Ramón y Cajal S/N, 28040 Madrid, Spain; 8Department of Biomedical Science and Human Oncology, University of Bari “Aldo Moro”, 70124 Bari, Italy; dr.falcicchio@gmail.com; 9Department of Woman, Child and of General and Specialized Surgery, University of Campania “Luigi Vanvitelli”, 80138 Naples, Italy; giusi.umano@gmail.com; 10Department of Neurosciences, ASST Carlo Poma, 46100 Mantova, Italy; dario.calafiore@gmail.com; 11Department of Medical and Surgical Sciences, University of Catanzaro “Magna Graecia”, 88100 Catanzaro, Italy

**Keywords:** periodontal disease, oral hygiene, vitamin D, vitamin D deficiency, preterm birth, low birth weight, pregnancy

## Abstract

Periodontal disease seems to be correlated with low vitamin D serum levels, preterm birth (PTB) and low birth weight (LBW), although the literature still lacks a consensus. This study aimed to investigate this correlation in a cohort of pregnant women over 20 weeks of gestation from the University Hospital “Maggiore della Carità”, Novara, Italy. We assessed serum levels of vitamin D and oral health status through the following indexes: Oral Hygiene Index (OHI), Plaque Control Record (PCR), Gingival Bleeding Index (GBI), and Community Periodontal Index of Treatment Needs (CPTIN). Moreover, we assessed the number of PTB and LBW among the newborns. Out of 121 pregnant women recruited, 72 (mean age 29.91 ± 3.64 years) were included. There was a statistically significant correlation between preterm and OHI > 3 (*p* = 0.033), and between LBW and OHI > 3 (*p* = 0.005) and CPITN = 3 (*p* = 0.027). Both pregnant women with vitamin D deficiency ((25-hydroxy-vitamin D) < 30 ng/mL) and PTB plus LBW newborns were significantly correlated (*p* < 0.05) with poor levels of all oral health status indexes during pregnancy. Furthermore, these conditions (women with hypovitaminosis D and combination of PTB and LBW) were shown to be significantly correlated (*p* < 0.001). Taken together, our findings reported a high prevalence of PTB and LBW with poor oral health and vitamin D deficiency in pregnant women.

## 1. Introduction

Periodontal disease is one of the more common inflammatory diseases in the adult population, with an incidence varying from 5 to 30% [1,2,3,4]. Periodontal disease, initiated by bacterial biofilms, can cause the destruction of soft and hard periodontal tissues, consequently leading to tooth loss [5,6,7]. Periodontal disease produces inflammatory mediators and microbial products that can enter the systemic circulation through the mouth and reach distant organs [8,9]. Indeed, several studies have shown a link between periodontal disease and some frequent systemic pathologies [10] such as diabetes [11], cardiovascular disease [12,13], respiratory disease [14], obesity [15], cancer [16,17], and preterm birth (PTB) [7]. Vitamin D is a secosteroid that can be ascribed to steroid hormones if considering its biological effects on several tissues [18,19,20,21,22]. Vitamin D might affect periodontal disease both through an effect on bone mineral density and through immunomodulatory effects [23,24,25]. Consequently, the effects of low vitamin D levels on periodontal health represent a matter of interest. Over the past few decades, vitamin D deficiency has been associated with gingivitis and periodontitis of different severities [26,27,28]. In vitro, studies demonstrated that vitamin D might decrease the number of *Porphyromonas gingivalis* (Pg) through active autophagy [29], and also the inflammatory burden of periodontitis in rodent models [30,31].

PTB (before 37 weeks of gestation) is a main cause of neonatal mortality worldwide and a major contributor to long-term disability among survivors [32]. Each year, about 15 million infants are born before the end of physiological pregnancy [33], and more than half of all permanent sequelae that infants suffer at cardiovascular, neurological, respiratory, and congenital levels could be attributed to PTB [18]. Moreover, PTB is associated with low birth weight (LBW) of less than 2500 grams, and, even with the great advances in obstetrics, the rate of LBW associated with PTB has not decreased along the last few decades [34].

Although the exact mechanisms by which periodontal disease could adversely affect pregnancy are still unclear, the presence of DNA of oral pathogens associated with periodontitis has been found in amniotic fluid [35,36], placental tissues [37], and the genital tract [38]. The anaerobic bacteria *Fusobacterium nucleatum* (Fn) and Pg are the most important periodontal pathogens that could induce placental inflammation and subsequent damage [39]. Particularly, Pg is the most common microorganism found in the amniotic fluid and placental tissue [40,41]. Bacteria involved in periodontal disease could spread from the oral cavity to the uterine district, causing the consequent cascade of immuno-inflammatory mediators such as PGE2, IL-6, IL-1, and TNF-alpha that might be implicated in adverse pregnancy outcomes [42,43,44,45]. Alternatively, inflammatory cytokines indirectly affect placental development and fetal growth leading to PTB due to their effects on uterine contractions [44,46,47,48]. 

Scientific evidence showed that due to the high heterogeneity of the investigated cohorts and due to the use of several non-clinical indices, the assessment of the prevalence of gingivitis during pregnancy is challenging [49,50,51], with wide ranges of reported values from 30 to 70% and up to 100% during the second and third trimester [44,52,53,54]. 

Moreover, hormonal changes related to pregnancy seem to play a role as a co-factor in periodontal disease onset [52,53,55], whereas chronic inflammation of deep periodontal tissue might negatively affect pregnancy outcomes [43,54,55,56,57,58,59,60,61,62], such as PTB and LBW.

Although a relationship between periodontal disease and PTB has been hypothesized [63,64,65], many confounding factors make it difficult to assess the relationship between these two conditions, as underlined by the lack of consensus within the scientific community [57,64,65,66,67,68].

Several studies have investigated the correlation among vitamin D deficiency, PTB, and LBW [69,70,71,72]. Recent studies have suggested that the effects of 25-hydroxy-vitamin D (25(OH)vit. D) on gestational weight gain might be explained by biologic activities of this micronutrient on adipose tissue. Indeed, vitamin D receptors are present on human adipocytes and 25(OH)vit. D levels seem to influence lipogenesis, lipolysis, adipogenesis, and reducing adipose tissue inflammation [73,74,75]. Additionally, due to the anabolic effect of 25(OH)vit. D on growth, vitamin D deficiency might be associated with impaired maternal weight gain and fetal growth among vitamin D-deficient mothers [76]. Moreover, a limited number of studies have suggested that 25(OH)vit. D concentration can be related to PTB and LBW [77,78], although the results of these studies are still debated and conflicting.

Hence, the present cross-sectional study aimed to increase the level of evidence on this controversial topic by investigating the correlation among hypovitaminosis D, periodontal disease, and PTB and LBW in a cohort of pregnant women.

## 2. Materials and Methods

### 2.1. Participants

In this cross-sectional study, we recruited all the adult pregnant women (i.e., >20 weeks of gestation) referred to the Obstetrics and Gynecology Unit of the University Hospital “Maggiore della Carità”, Novara, Italy, over a 12-month period (January 2019—December 2019). Exclusion criteria were: (1) psychiatric or neurological diseases; (2) evidence of main concurrent diseases; (3) undergoing treatment with corticosteroids, immunoglobulin, or immunosuppressive drugs; (4) fully edentulous patients; (5) suffering or having suffered in the past from major oral infections; (6) being unable to understand the informed consent; (7) patients undergoing vitamin D supplementation.

This study was approved by the local ethics committee (CE 61/10, prot.392). All participants were asked to carefully read and sign an informed consent form, and researchers protected the privacy and the study procedures according to the Declaration of Helsinki with pertinent national and international regulatory requirements. Data collection and reporting were performed in accordance with the STrengthening the Reporting of OBservational studies in Epidemiology (STROBE) Guidelines.

### 2.2. Clinical Assessment and Outcome Measures

Medical history of all patients was collected considering the following data: age, ethnicity, body mass index (BMI) before pregnancy, health condition (hyperglycemia, hypertension, depressive syndrome, poor weight gain, nephropathy, broncho pneumopathies), and alcohol, smoke, and narcotic consumption. In addition, specific obstetric and gynecologic information was collected, such as: number and outcome of previous pregnancies, type of previous birth, any obstetric complication of previous pregnancies (such as gestational hypertension and preeclampsia, urinary tract infection, vaginal bleeding, miscarriage, PTB, postpartum depression), characteristics of pregnancies, and birth weight of the newborns. Lastly, we assessed serum 25(OH)vit. D (ng/mL) and the number of patients with serum levels of 25(OH)vit. D < 30 ng/mL and <20 ng/mL.

Finally, all the participants underwent an oral clinical examination at the Dental Clinic of the University Hospital “Maggiore della Carità”, Novara, Italy. The clinical examination was performed by M.M., a dentist with more than 30 years of experience in periodontal diagnosis and therapy. The examination included the recording of the following outcomes: simplified Oral Hygiene Index (OHI) [79], for the presence of debris/stain and tartar on the dental elements; Plaque Control Record (PCR) [80], to assess the presence of plaque on the dental elements; Gingival Bleeding Index (GBI) [81,82], to evaluate gingival inflammation; and Community Periodontal Index of Treatment Needs (CPTIN) [83], to assess periodontal treatment needs.

### 2.3. Statistical Analysis

Data management and analyses were conducted according to a pre-specified statistical analytical plan. Statistical analysis was performed using STATA v. 12 (StataCorp LP, College Station, TX, USA). The continuous variables are presented as means ± standard deviations, or median and interquartile range. The Shapiro–Wilk test was performed to assess the distribution of all continuous data; as the data did not follow a normal distribution, Wilcoxon rank sum test was used to compare continuous variables between the two groups. Fisher exact test was performed to compare categorical variables between the two groups. Pearson correlation coefficients and regression analyses assessed associations and correlations among the oral health status of study participants, analyzing a correlation with clinical and demographic features. A *p*-value of 0.05 was considered statistically significant.

## 3. Results

From the 121 patients recruited, 72 women, mean age 29.91 ± 3.64 years, fulfilling the inclusion criteria, underwent oral clinical examination and were enrolled in the present study. Fifty-two women were Caucasian (72.2%), fourteen African (19.4%), and six Asian (8.3%). Gestational age was between the 20th and 25th week for twenty-one women, twenty-seven were between the 25th and 30th week, twenty-two were between the 30th and 35th week, and only two were over the 35th week. We identified six underweight women (body mass index, BMI ≤ 18.49) and fifty-five had BMI > 18.49 and <24.99. Ten women were overweight (BMI ≥ 25.00) and one was obese (BMI ≥ 30) (Figure 1 depicts a pregnant woman with periodontal disease).

Five women referred to complications of their previous pregnancy and five had PTB. One woman treated her hyperglycemia with an oral hypoglycemic. Another one was affected by thrombophilia. Only one had a story of uterine myomas. Uro-genital inflammation was diagnosed in six women before the ongoing pregnancy. Four women underwent uterine surgery. One pregnancy was medically assisted, while seventy-one were spontaneous. Details of patients’ characteristics are reported in Table 1.

The mean serum level of 25(OH)vit. D was 20.5 ± 6.7 ng/mL; fifty-nine patients (90.3%) reported hypovitaminosis D ((25(OH)vit. D) < 30 ng/mL) and 32 (44.4%) had a serum level of (25(OH)vit. D) < 20 ng/mL. On gingival probing, GBI was >25 in forty patients and the OHI was unsatisfactory due to plaque, pigmentation, and calculus in twenty-seven women. With respect to the CPTIN score, fifty-nine women required periodontal care. Patients’ periodontal conditions are listed in Table 2.

No miscarriages were recorded among all the women studied. Sixty women had a vaginal delivery while cesarean section was performed in twelve cases. Fourteen of these were twin pregnancies. Fifty-one women (70.83%) had full-term delivery (after 37th week of gestation), 20 women (27.78%) had late preterm delivery (between 32nd and 37th week), and only one patient had early preterm delivery (between 28th and 32nd week). Fifty-six neonates (77.8%) had normal weight at birth (>2500 < 4000 g), 14 neonates (19.4%) were low-weight (≥1501 ≤ 2500 g) and 2 (2.8%) were macrosomic (≥4000 g). A large part of the population (*n* = 62, 86.11%) gave birth at term to normal weight or macrosomic neonates, while 13.89% (*n* = 10) had PTB and low-weight neonates.

The statistical analysis did not show a significant correlation (*p* > 0.05) among preterm delivery and the value of PCR, GBI, and CPTIN. On the other hand, there was a statistically significant correlation between preterm and OHI>3 (*p* = 0.033). Similarly, no statistically significant correlation was detected in these groups between LBW and PCR and GBI values (*p* > 0.05), while there was a significant correlation in the case of OHI>3 (*p* = 0.005) and CPITN = 3 (*p* = 0.027).

Lastly, considering the combination of PTB and LBW, we found a significant correlation with OHI>3 (*p* = 0.001), PCR>75% (*p* = 0.035), GBI≥50% and <75% (*p* = 0.035), and CPTIN = 3 (*p* < 0.004).

Concerning the serum level of vitamin D, a statistically significant correlation was found between vitamin D deficiency (serum level of 25(OH)vit. D) < 30 ng/ml) and PCR>75% (*p* = 0.021), GBI>50% (*p* = 0.002), OHI>3 (*p* < 0.008), and CPTIN code 3 (*p* < 0.014). Furthermore, women with hypovitaminosis D have a significant correlation with the combination of PTB and LBW (*p* < 0.001).

## 4. Discussion

The aim of the present cross-sectional study was to investigate the correlation among hypovitaminosis D, periodontal disease, PTB, and LBW in a cohort of pregnant women.

Based on the presented data, a significant correlation among the birth of premature babies with OHI>3 as well a correlation between the birth of low-weight neonates and both OHI>3 and code 3 of the CPTIN could be detected. Interestingly, by correlating the birth of both premature and underweight neonates with the dento-periodontal indices, we obtained statistically significant correlations with them.

The prevalence of vitamin D deficiency in our study sample was very high (90.3%), and a statistically significant correlation was found between vitamin D deficiency and all the oral status indexes. Furthermore, women with hypovitaminosis D have a significant correlation with the combination of PTB and LBW (*p* < 0.001), although there was only a non-significant trend with the two parameters (PTB and LBW), considered as single variables.

The potential mechanisms of how periodontal disease could affect pregnancy outcomes are still not totally understood [68]. Nevertheless, it must be underlined that periodontal infection induces elevated cytokine levels in the host [84,85], and that elevated serum and/or amniotic fluid levels of proinflammatory cytokines, such as IL-1, IL-6, and TNF-α, may stimulate the production of prostaglandins in the chorion, which is associated with intra-amniotic inflammation and PTB development [86,87]. In addition, the hematogenous dissemination of periodontal pathogens could trigger metastatic infection at the feto-placental unit. Moreover, during pregnancy, the elevated levels of female sex hormones are responsible for increasing vascular permeability, which has been clinically correlated to an increased level of oral plaque and consequently gum bleeding scores [68].

Studies showed that periodontal pathogens can be detected in the maternofetal unit, where they might be involved in the development and progression of inflammation [40,88]. Moreover, there is striking evidence for an increased risk of PTB through vaginal infections, as well as for an association between the oral and the vaginal microbiome (e.g., low hygiene status is associated with increased bv risk) [89,90]. Indeed, an antenatal infection screen-and-treat program might be useful in routine pregnancy care to prevent PTB, LBW, and adverse pregnancy outcomes [89,91].

Furthermore, it was interesting to notice the strong correlation among hypovitaminosis D and periodontal disease, even if in the literature the results are controversial. In a recent metanalysis, Machado et al. [92] supported an association between serum vitamin D levels and chronic periodontitis. On other hand, a review by Miller and Pavlesen [93] investigated the relationship between vitamin D and chronic periodontal disease in older adults, affirming that, although the biologic mechanisms suggest that vitamin D deficiency could be a risk factor for the development or progression of periodontal disease, the findings from studies were overly supportive. With respect to our results, they are confirmatory of these previous findings.

Moreover, in our study, a similar positive correlation between vitamin D insufficiency and PTB with LBW was found. Wang et al. [94] affirmed that 25(OH)vit. D sufficiency in the second and third trimesters was associated with higher gestational duration. Similar results were shown in a recent metanalysis on 24 studies [95]. The authors reported that vitamin D deficiency in the second trimester is likely associated with an increased risk of PTB with an odds ratio of 1.12. These findings were confirmed by a systematic review [96], suggesting that women receiving vitamin D supplementation during pregnancy not only had improved maternal vitamin D levels at term, but also a reduced risk of LBW (average risk ratio 0.40).

Therefore, an oral hygiene assessment should always be taken into consideration in pregnant women in common clinical practice, as has also been performed for other disabling conditions [28,97,98]. In this scenario, we recommend that periodontal therapy and oral rehabilitation should be included within the clinical management of the oral health status in pregnant women.

Moreover, the elevated demand for micronutrients during pregnancy leads to women being at high risk of vitamin D deficiency and periodontal disease; thus, adequate vitamin D supplementation plays a crucial role in pregnant women to reduce the risk of periodontal disease, and even the risk of PTB and LBW in newborns [95,99,100].

This study is not free from limitations: first, the cross-sectional study design precludes it from external validity; second, even though all the patients referred to the obstetric clinic were screened and potentially eligible for the study, the dataset included a small sample. In addition, the decision to use different periodontal screening indices to evaluate oral health status was taken to increase the feasibility of the study; however, we are aware that a complete periodontal chart (with full-mouth plaque and full-mouth bleeding values), as well as radiographic and microbiological analysis, would have provided additional information for this study.

## 5. Conclusions

Taken together, in this cross-sectional study, we assessed the correlation among hypovitaminosis D, periodontal disease, PTB, and LBW in a cohort of pregnant women. We showed that a high prevalence of PTB and LBW is present in pregnant women affected by periodontal disease and vitamin D deficiency. In light of these results, an adequate screening of oral health and 25(OH)vit. D serum levels should be implemented in common clinical practice for pregnant women. Further prospective studies on larger cohorts are warranted to investigate the role that oral health rehabilitative programs and vitamin D supplementation might have in reducing periodontal disease in pregnant women and even the incidence of PTB and LBW.

## Figures and Tables

**Figure 1 jcm-10-04578-f001:**
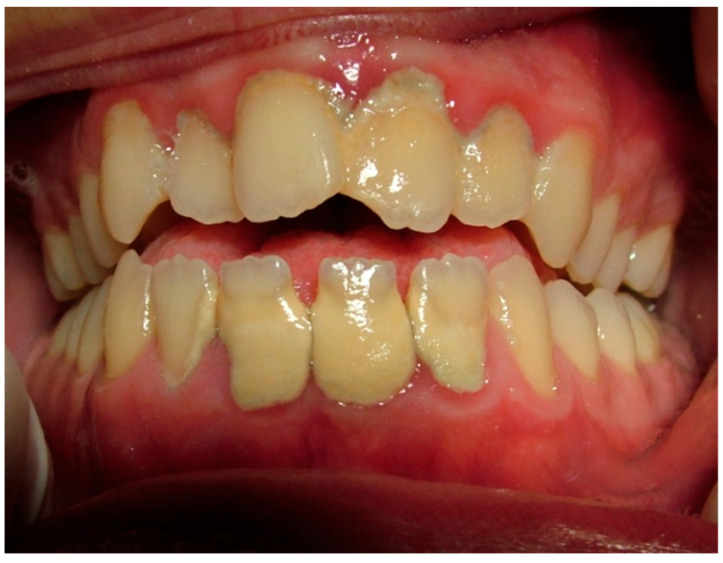
A case of a pregnant woman with periodontal disease due to poor oral health status.

**Table 1 jcm-10-04578-t001:** Anamnestic and clinical characteristics of the study participants (*n* = 72).

Age (Years)	29.91 ± 3.64
Gestational age between 20th and 25th week	21 (29.2%)
Gestational age between 25th and 30th week	27 (37.5%)
Gestational age between 30th and 35th week	22 (30.6%)
Gestational age over 35th week	2 (2.8%)
Previous pregnancy	36 (50.0%)
- one full-term pregnancy	25 (69.4%)
- between 2 and 3 full-term pregnancies	9 (12.5%)
- more than 3 full-term pregnancies	2 (5.6%)
- complications	5 (13.9%)
- miscarriages	13 (36.1%)
- single pregnancies	29 (80.6%)
- twin pregnancies	7 (19.4%)
- natural birth	30 (83.3%)
- caesarean section	6 (16.7%)
- preterm birth	5 (13.9%)
PBLW risk factors	40 (55.6%)
Smokers	7 (9.7%)
Drink alcohol	0 (0%)
Supplementation with fluorides	4 (15.4%)
(25(OH)vit. D) serum levels	20.51 ± 6.7
25(OH)vit. D < 30 ng/mL	65 (90.3%)
25(OH)vit. D < 20 ng/mL	32 (44.4%)

Continuous variables are expressed as means ± standard deviations; categorical variables are expressed as counts/percentages. Abbreviations: PBLW= preterm birth plus low weight; 25(OH)vit. D = 25-hydroxy-vitamin D.

**Table 2 jcm-10-04578-t002:** Oral hygiene status in pregnant women included in this study (*n* = 72).

	Total (*n* = 72)	PTB (*n* = 21)	LWB (*n* = 14)	PBLW (*n* = 10)
≤1.2	45 (62.5%)	7 (33.3%)	3 (21.4%)	2 (20.0%)
>1.2 ≤ 3.0	20 (27.8%)	5 (23.8%)	3 (21.4%)	3 (30.0%)
>3.0 ≤ 6.0	7 (9.7%)	9 (42.9%)	8 (57.1%)	5 (50.0%)
PCR				
From 0% to 25%				
>25% and ≤50%	23 (31.9 %)	3 (14.3%)	2 (14.3%)	0 (0%)
>50% and ≤75%	17 (23.6%)	10 (47.6%)	5 (35.7%)	1 (10.0%)
>75%	14 (19.4%)	5 (23.8%)	5 (35.7%)	7 (70.0%)
GBI				
From 0% to 25%				
>25% and ≤50%	13 (18.1%)	11 (52.4%)	8 (57.1%)	3 (30.0%)
>50% and ≤75%	24 (13.3%)	0 (0%)	3 (21.4%)	4 (40.0%)
>75%	3 (4.2%)	0 (0%)	3 (21.4%)	3 (30.0%)
CPTIN				
0				
1	19 (26.4%)	4 (19.1%)	0 (0%)	0 (0%)
2	23 (31.9%)	8 (38.1%)	6 (42.9%)	2 (20.0%)
3	17 (23.6%)	9 (42.9%)	8 (57.1%)	8 (80.0%)
≤1.2	45 (62.5%)	7 (33.3%)	3 (21.4%)	2 (20.0%)

Categorical data are expressed as counts (%). Abbreviations: PTB = preterm birth; LWB = low-weight birth; PBLW = preterm birth plus low-weight birth; OHI = Oral Hygiene Index; PCR = Plaque Control and Record; GBI = Gingival Bleeding Index; CPTIN = Community Periodontal Index of Treatment Needs.

## Data Availability

The dataset is available upon request.

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
