# Peer review of "Periodontal Disease and Vitamin D Deficiency in Pregnant Women: Which Correlation with Preterm and Low-Weight Birth?"

_jcm, 2021, doi:10.3390/jcm10194578_

Round 1

Reviewer 1 Report

I hope to consider the following comments:

Title:

  • It is reflecting the aim and the paper contents so succinctly.

Abstarct:

  • Abstract is clear and provides good summary of the manuscript contents.
  • Would you remove ‘the’ from the following sentence: “This cross-sectional study aimed to the investigate this correlation in a cohort of pregnant women over 20 weeks of gestation.” I advise to be “This cross-sectional study aimed to investigate this correlation in a cohort of pregnant women over 20 weeks of gestation.” 
  • In the abstract you need to specify the place (location and/or country) where the study was conducted?

Keywords:

  • The keywords are representative for the manuscript content.

Background:

  • The background is well structured and followed a logical order. The introduction describes the extent of periodontal diseases problem among adult population, its causes and its potential complications. In addition, the authors provide a good and thorough review for how the deficiency of Vitamin D could correlate to periodontal diseases, and subsequently, how the deficiency of Vitamin D and periodontitis may negatively affect Preterm birth and birth weight. Afterward, the authors identify the gap in knowledge and the importance of the proposed study.  
  • The aim of the study is clearly defined and consistent with the rest of the manuscript.
  • The citation used are intensive, relevant to authors’ work, up to date and provide supportive background for the article.

Methods:

  • Methods are described in detail. Materials and instruments used for data collection, and ethical approvals are explained clearly. The outcome and analyses methods are well defined and sound.
  • But information is required to explain how the authors tested the validity of the questionnaire completed by participants? e.g Was there a pilot study to test questionnaires’ validity?

Results:

  • The figures and tables in the manuscript accurately describe the results.
  • This section highlights the most important findings and directly answers the main question and the aim of the study.

Discussion and conclusion:

  • It provides a good interpretation and explanation of the study results and its implications. The discussion section builds a good evidence, which was highly supportive for the paper aims and conclusion. Furthermore, it provides in-depth discussion for the study limitation.
  • The conclusion is well written and highlighted future suitable programme.

References:

  • All citations in the manuscript appeared in the reference list.

Author Response

I hope to consider the following comments:

Title:

It is reflecting the aim and the paper contents so succinctly.

We would like to thank the reviewer for the comment. We are glad that she/he appreciated the title.

Abstarct:

Abstract is clear and provides good summary of the manuscript contents.

We would like to thank the reviewer for the comment. We are glad that she/he appreciated the Abstract

Would you remove ‘the’ from the following sentence: “This cross-sectional study aimed to the investigate this correlation in a cohort of pregnant women over 20 weeks of gestation.”

I advise to be “This cross-sectional study aimed to investigate this correlation in a cohort of pregnant women over 20 weeks of gestation.”

Thank you. We corrected the typo.

In the abstract you need to specify the place (location and/or country) where the study was conducted?

Thank you for the suggestion. We added the name of the Hospital and Country.  

Keywords:

The keywords are representative for the manuscript content.

Thank you.

Background:

The background is well structured and followed a logical order. The introduction describes the extent of periodontal diseases problem among adult population, its causes and its potential complications. In addition, the authors provide a good and thorough review for how the deficiency of Vitamin D could correlate to periodontal diseases, and subsequently, how the deficiency of Vitamin D and periodontitis may negatively affect Preterm birth and birth weight. Afterward, the authors identify the gap in knowledge and the importance of the proposed study. 

The aim of the study is clearly defined and consistent with the rest of the manuscript.

The citation used are intensive, relevant to authors’ work, up to date and provide supportive background for the article.

We would like to thank the reviewer for the comment. We are glad that she/he appreciated the Introduction.

Methods:

Methods are described in detail. Materials and instruments used for data collection, and ethical approvals are explained clearly. The outcome and analyses methods are well defined and sound.

We would like to thank the reviewer for the comment. We are glad that she/he appreciated the Methods.

But information is required to explain how the authors tested the validity of the questionnaire completed by participants? e.g Was there a pilot study to test questionnaires’ validity?

We would like to thank the reviewer for the comment. We have inappropriately defined as questionnaire the questions regarding the medical history of our patients. We corrected that part, accordingly.

Results:

The figures and tables in the manuscript accurately describe the results.

This section highlights the most important findings and directly answers the main question and the aim of the study.

We would like to thank the reviewer for the comment.  

Discussion and conclusion:

It provides a good interpretation and explanation of the study results and its implications. The discussion section builds a good evidence, which was highly supportive for the paper aims and conclusion. Furthermore, it provides in-depth discussion for the study limitation.

The conclusion is well written and highlighted future suitable programme.

We would like to thank the reviewer for the comment.   

References:

All citations in the manuscript appeared in the reference list.

We would like to thank the reviewer for the comment.  

Reviewer 2 Report

Interesting paper in a field with limited clinical data

Minor revisions are needed (in yellow into text)

Author Response

Interesting paper in a field with limited clinical data

Minor revisions are needed (in yellow into text)

We would like to thank the reviewer for the comment. We are glad that she/he appreciated the work. We modified the text according to her/his suggestions.